# META-LEARNING FROM DEMONSTRATIONS IMPROVES COMPOSITIONAL GENERALIZATION

## ABSTRACT

We study the problem of compositional generalization of language-instructed agents in gSCAN. gSCAN is a popular benchmark which requires an agent to generalize to instructions containing novel combinations of words, which are not seen in the training data. We propose to improve the agent's generalization capabilities with an architecture inspired by the Meta-Sequence-to-Sequence learning approach (Lake, 2019). The agent receives as a context a few examples of pairs of instructions and action trajectories in a given instance of the environment (a support set) and it is tasked to predict an action sequence for a query instruction for the same environment instance. The context is generated by an oracle and the instructions come from the same distribution as seen in the training data. In each training episode, we also shuffle the indices of the actions and the words of the instructions to make the agent figure out the relations between the actions and the words from the context. Our predictive model has the standard Transformer architecture. We show that the proposed architecture can significantly improve the generalization capabilities of the agent on one of the most difficult gSCAN splits: the "adverb-to-verb" Split H.

## 1 INTRODUCTION

We want autonomous agents to have the same compositional understanding of language that humans do (Chomsky, 1957; Tenenbaum, 2018). Without this understanding, the sample complexity required to train them for a wide range of compositions of instructions would be very high (Sodhani et al., 2021; Jang et al., 2021). Naturally, such compositional generalization has received interest from both the language and reinforcement learning communities. "Compositional Generalization" can be divided into several different sub-skills, for example being able to reason about object properties compositionally (Chaplot et al., 2018; Qiu et al., 2021), composing sub-instructions into a sequence (Logeswaran et al., 2022; Min et al., 2021) or generating novel action sequences according to novel instructions made up of familiar components (Lake & Baroni, 2018). In this work, we examine this latter challenge in more detail using the gSCAN environment (Ruis et al., 2020).

gSCAN is a testing environment for language-grounded agents, consisting of a 6-by-6 grid-world where each episode has a unique combination of objects, an initial agent position (in this work, the *state*) and some language instruction. The objects are one of a circle, cylinder or triangle, can be five different sizes and come in the colors red, blue, purple and yellow. We encode each cell in the state is encoded as bag of words, similar to BabyAI (Chevalier-Boisvert et al., 2019). The components are the object size, color, shape, and possible agent occupancy and direction. Available instruction action words are `push`, `pull` and `walk to` and available instruction adverbs are `while spinning`, `while zigzagging`, `hesitantly` and `cautiously`. Actions which the agent must produce are in the set of `WALK`, `STAY`, `LTURN`, `RTURN`, `PUSH` and `PULL`. In this work, when we refer to an action that is repeated more than once, we place the repeat count in parenthesis, like `LTURN(4)`. A *success* happens when the agent exactly matches the target actions for an episode.

The instructions follow a template of "`action a size? color? object adverb`", where `?` indicates that a token is optional. Certain combinations of instructions, actions, objects and adverbs are not found in the training set. They are instead found in certain held out "Splits". Split A is an in-distribution validation set split, containing instructions, target objects and target locations which can be found in the training set. Splits B to F are out-of-distribution sets where the target object has

an unseen description made up of combinations of familiar terms (for example red square in Split C) or is at a location not seen before during training (for example, southwest of the agent in Split D). These test splits, except Split D, have been solved by a using a Transformer (Qiu et al., 2021). Split G is a "meta-learning" split, where an example of "cautiously" is seen only $k$ times during training. Split H, otherwise known as the "adverb-to-verb" split, contains only instructions following the template "`pull a size? color? object while spinning`". This requires the agent to walk towards the target object and pull it the required number of times, while at the same time performing actions `LTURN(4)` after each `WALK` and `PULL`. Solving the problem requires the agent to generate the unseen action trajectory based on a compositional understanding of the instructions. In this work, we mainly focus on Split H.

We hypothesize that a promising approach is Meta Sequence-to-Sequence Learning (meta-seq2seq) proposed by Lake (2019). We think the reason why this approach works well is the permutations applied to the supports and target instructions, which ensures that an agent does not overfit to particular sequences of symbols in the output space and instead forces meta-learning to determine what the true output actions should be for a given episode. Extending this approach to language grounding environments was flagged as a possible future work direction by Ruis et al. (2020). In this work we propose to do exactly that.

Our contributions are: **first**, we describe an extension of meta-seq2seq with state-relevant supports, **second** we report promising success rate performance on gSCAN Split H, **third** we demonstrate the importance of generating relevant supports and show how different procedures for generating supports affect performance, and **fourth** we motivate how this approach aligns with intuitions about human compositional problem solving.

## 2 RELATED WORK

**Compositional Generalization**  There is a long line of work on the challenge of compositional generalization in deep learning. Initial works show that sequence models such as RNNs cannot solve these problems well (Tenenbaum, 2018; Loula et al., 2018). Datasets such as SCAN (Lake & Baroni, 2018), 0gendata (Geiger et al., 2019), COGS (Kim & Linzen, 2020), and PCFG (Hupkes et al., 2020) serve as benchmarks to measure progress. Various approaches have been proposed to improve compositional generalization performance, including data augmentation (Andreas, 2020; Shi et al., 2021; Guo et al., 2020a; Qiu et al., 2022), problem-specific inductive biases (Russin et al., 2020; Guo et al., 2020b; Yin et al., 2021; Chakravarthy et al., 2022; Spilsbury & Ilin, 2022), increased data diversity (Patel et al., 2022; Andreas, 2020), transfer learning (Zhu et al., 2021) and modular networks (Andreas et al., 2016; Ruis & Lake, 2022). These approaches can perform very well, but usually require prior assumptions about underlying data. In computer vision and multimodal domains, the Transformer architecture has been shown to solve some compositional generalization tasks (Vaswani et al., 2017; Hudson & Zitnick, 2021; Chen et al., 2020). Transformer's success on token-level tasks is also promising but still limited (Power et al., 2022; Bhattamishra et al., 2020; Qiu et al., 2021; 2022; Sikarwar et al., 2022). Meta-learning (Conklin et al., 2021; Yang et al., 2022; Mitchell et al., 2021) and group equivariance (Gordon et al., 2020) have also shown promise on such problems.

**Grounded Environments**  Many language grounding environments exist, such as BabyAI (Chevalier-Boisvert et al., 2019), ALFRED (Shridhar et al., 2020), VizDoom (Chaplot et al., 2018) and SILG (Zhong et al., 2021). gSCAN and its derivatives (Ruis et al., 2020; Wu et al., 2021) specifically focus on task compositional generalization in an interactive world. The various splits of gSCAN are still not completely solved. Various approaches including graph networks (Gao et al., 2020), linguistic-assisted attention (Kuo et al., 2021), symbolic reasoning Nye et al. (2021), auxiliary tasks Jiang & Bansal (2021) Transformers (Qiu et al., 2021), modular networks (Heinze-Deml & Bouchacourt, 2020; Ruis & Lake, 2022), data augmentation (Setzler et al., 2022; Ruis & Lake, 2022) have been proposed. Splits D, G and H remain challenging to solve with a general approach.

**In-context and model-based learning**  We take inspiration from a long line of work on in-context meta-learning, starting with $RL^2$ (Duan et al., 2016) for RNNs and the extension to Transformers with TrML (Melo, 2022). Also related are the ideas of retrieval for in-context learning (Goyal et al., 2022; Borgeaud et al., 2022) and proposing goals and planning in an imagination of the world (Nair et al., 2018; Chane-Sane et al., 2021; Hafner et al., 2020; Deac et al., 2021).

## 3 METHOD

### 3.1 MOTIVATION

Many existing solutions to this problem build an autoregressive model of the generated target sequences. Our hypothesis is that those models overfit to a particular distribution of actions seen in the training data.

The nature of the problem can be seen in the one-step-ahead action conditional probability distributions in Table 1. While these one-step-ahead conditional probabilities do not capture the full picture, since there are other variables such as other previous actions as well as the instruction, they do give a snapshot of the nature of the distribution shift problem in Split H. In the training split, both $P(\text{LTURN}|\text{PULL}) = 0$ and $P(\text{PULL}|\text{LTURN}) = 0$. On the contrary, and unlike the other the test splits of gSCAN (see for example the "red squares" split C), $P(\text{PULL}|\text{LTURN}) = 1$ and $P(\text{LTURN}|\text{PULL}) > 0$ in Split H.

Table 1: One-step ahead conditional distributions in various gSCAN splits for targets. Shown is $P(a_t|a_{t-1})$ where $a_t$ is down the rows and $a_{t-1}$ is along the columns. Bolded are conditional probabilities notably different from the training data.

(a) Training data

|       | PULL | PUSH | STAY | LTURN | RTURN | WALK |
|-------|------|------|------|-------|-------|------|
| PULL  | 0.52 | 0    | 0.33 | 0     | 0     | 0.15 |
| PUSH  | 0    | 0.27 | 0.26 | 0.26  | 0     | 0.21 |
| STAY  | 0.19 | 0.05 | 0    | 0     | 0     | 0.75 |
| LTURN | 0    | 0.01 | 0.01 | 0.73  | 0     | 0.24 |
| RTURN | 0    | 0    | 0.18 | 0.14  | 0     | 0.68 |
| WALK  | 0    | 0    | 0.20 | 0.35  | 0.17  | 0.28 |

(b) Test Split A

|       | PULL | PUSH | STAY | LTURN | RTURN | WALK |
|-------|------|------|------|-------|-------|------|
| PULL  | 0.52 | 0    | 0.32 | 0     | 0     | 0.16 |
| PUSH  | 0    | 0.28 | 0.25 | 0.27  | 0     | 0.21 |
| STAY  | 0.25 | 0.07 | 0    | 0     | 0     | 0.68 |
| LTURN | 0    | 0.02 | 0.02 | 0.73  | 0     | 0.23 |
| RTURN | 0    | 0    | 0.20 | 0.13  | 0     | 0.67 |
| WALK  | 0    | 0    | 0.19 | 0.33  | 0.19  | 0.29 |

(c) Test Split C

|       | PULL | PUSH | STAY | LTURN | RTURN | WALK |
|-------|------|------|------|-------|-------|------|
| PULL  | 0.51 | 0.00 | 0.33 | 0.00  | 0.00  | 0.15 |
| PUSH  | 0.00 | 0.23 | 0.26 | 0.26  | 0.00  | 0.25 |
| STAY  | 0.20 | 0.04 | 0.00 | 0.00  | 0.00  | 0.76 |
| LTURN | 0.00 | 0.01 | 0.01 | 0.73  | 0.00  | 0.25 |
| RTURN | 0.00 | 0.00 | 0.18 | 0.14  | 0.00  | 0.69 |
| WALK  | 0.00 | 0.00 | 0.20 | 0.34  | 0.17  | 0.28 |

(d) Test Split H

|       | PULL | PUSH | STAY | LTURN | RTURN | WALK |
|-------|------|------|------|-------|-------|------|
| PULL  | 0    | 0    | 0    | **1.00** | 0  | 0    |
| PUSH  | 0    | 0    | 0    | 0     | 0     | 0    |
| STAY  | 0    | 0    | 0    | 0     | 0     | 0    |
| LTURN | **0.08** | 0 | 0 | 0.78  | 0     | 0.14 |
| RTURN | 0    | 0    | 0    | 1.00  | 0     | 0    |
| WALK  | 0    | 0    | 0    | 0.89  | 0.11  | 0    |

This poses a problem for any model conditioned on past actions, because giving any probability mass to a token where $P(a_t|a_{t-1}) = 0$ in the training data will result in some error according to the cross-entropy loss function. At the same time, the gSCAN problem as a whole calls for some level of timestep-dependent behaviour, so modelling $P(a_t|a_{t-1})$ is required. We believe that any approach which performs well on these two splits will need to handle this problem in some way.

### 3.2 META-SEQUENCE-TO-SEQUENCE LEARNING

Meta Sequence-to-Sequence learning (meta-seq2seq, Lake (2019)) is type of meta-learning introduced to handle certain classes of sequence-to-sequence compostional generalization problems like the one described in Section 3.1. In a *few-shot* or *meta-learning* problem context, the objective is to use some examples of a particular class of problem called *supports* in order to estimate the *target* for some *query*. Better known examples of this framework include MAML Finn et al. (2017), RL[2] (Duan et al., 2016) and TrML (Melo, 2022). The supports consist of *support instructions* $I_1, ..., I_n$ and corresponding *support targets* $A_1, ..., A_n$ (in this case the actions). A *query instruction* $I^Q$ is given and the model must predict the corresponding *query targets* $I^A$. The critical part of the framework is that the token-symbol mapping for both the *supports* and *query* inputs and targets are re-permuted on every episode. A consistent permutation is applied for both the queries and supports. Typically in sequence-modelling problems, a tokenizer or dictionary is used to map words or tokens to integer symbols. The model operates at the level of symbols, then the user typically translates the output back into tokens. The main idea behind meta-seq2seq is learn a model that can handle *permutations* in the token-symbol mapping by providing *supports* of how a given permutation of symbols are used to solve other related problems. The permutations make the task unsolvable without the supports, forcing the model to make use of them and also not to overfit to particular

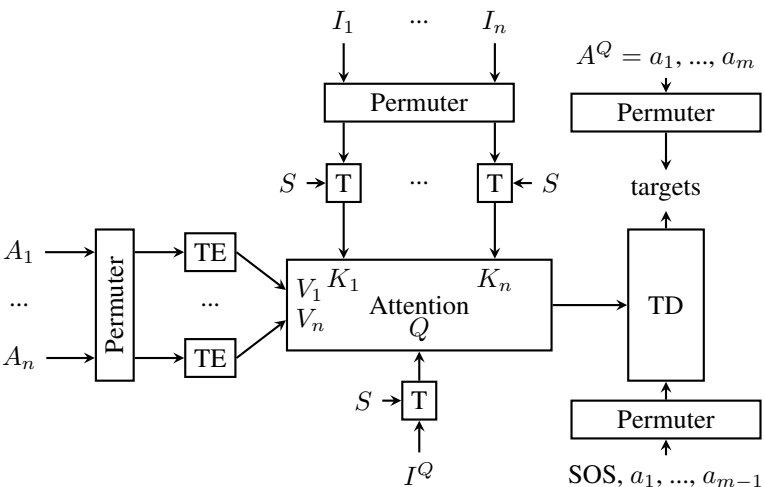

Figure 1: A schematic showing our proposed extension of meta-seq2seq (Lake, 2019). For a given episode with the initial state $S$, an oracle function generates state-relevant in-distribution support instructions sequences $I_1, ..., I_n$, which contain parts of the query instruction, but not the query itself. The action generator, which could be an oracle function or an existing in-distribution model, generates the corresponding support target sequences $A_1, ..., A_n$ for those instructions in the environment. Permuter blocks shuffle the token-symbol indices for both the instructions and targets on each episode. $S$ is encoded and each of $I_1, ..., I_n$ are decoded using a Transformer Encoder-Decoder (T), summarized as a vector by using a [CLS] token in the decoder sequence. $A_1, ..., A_n$ are encoded into vectors in the same way using a Transformer Encoder (without $S$). $I^Q$ and $S$ are encoded using the same T, except that the decoded $I^Q$ sequence is taken as the output as opposed to the decoded [CLS] token. Attention between each decoded symbol in $I^Q$ in and each decoded support instruction vector $I_1, ..., I_n$ provides mixing weights for each of the encoded support target vectors in $A_1, ..., A_n$, resulting in an output sequence the same length as $I^Q$. A Transformer Decoder (TD) takes right-shifted causally masked ground truth permuted targets as input and the Attention block output as context and predicts the permuted targets. At testing time, no permutation is applied.

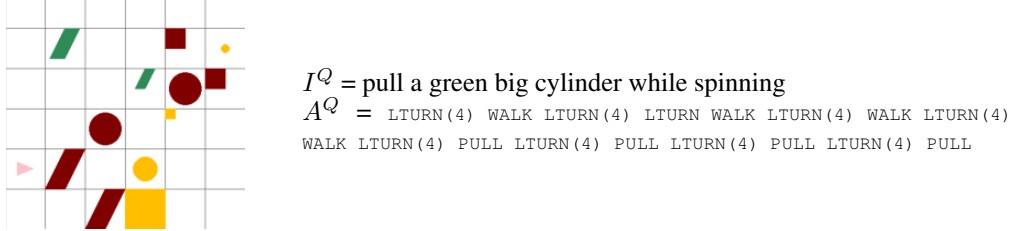

$I^Q$ = pull a green big cylinder while spinning
$A^Q$ = `LTURN(4) WALK LTURN(4) LTURN WALK LTURN(4) WALK LTURN(4)`
`WALK LTURN(4) PULL LTURN(4) PULL LTURN(4) PULL LTURN(4) PULL`

$I_1$ = walk to a green big cylinder while spinning
$A_1$ = `LTURN(4) WALK LTURN(4) LTURN WALK LTURN(4) WALK LTURN(4) WALK`
$I_2$ = push a green big cylinder while spinning
$A_2$ = `LTURN(4) WALK LTURN(4) LTURN WALK LTURN(4) WALK LTURN(4) WALK`
`LTURN(4) PUSH LTURN(4) PUSH`
$I_3$ = pull a green big cylinder while zigzagging
$A_3$ = `WALK LTURN WALK WALK WALK PULL PULL PULL PULL`
$I_4$ = pull a green big cylinder hesitantly
$A_4$ = `WALK STAY LTURN WALK STAY WALK STAY WALK STAY PULL STAY PULL`
`STAY PULL STAY PULL STAY`
$I_5$ = pull a green big cylinder
$A_5$ = `WALK LTURN WALK WALK WALK PULL PULL PULL PULL`

Figure 2: Example of the context generated by the oracle for one episode of training. Above: State $S$, query instruction $I^Q$ and target action sequence $A^Q$. Below: Support instructions $I_1, ..., I_5$ and action sequences $A_1, ..., A_5$.

symbol sequences. In the meta-seq2seq framework, the permutations are applied using a Permuter block during the episode generation process (see Appendix E for examples of permutations). At testing time, no permutations are applied. The meta-seq2seq architecture encodes each $I_1, ..., I_n$ and each query symbol in $I^Q$ computes attention between the two, then uses attention between $I^Q$ and $I_1, ..., I_n$ to mix the representations for each encoded support target $A_1, ..., A_n$ into a sequence of the same length as $I^Q$, which is the input sequence to a decoder used to predict the *query targets*.

Meta-seq2seq differs from other contemporary approaches to compositional generalization in language such as "Good Enough Compositional Augmentation" (GECA, Andreas (2020)). GECA is an augmentation method which recognizes *template fragments* in text, then realizes those templates with other possible substitutions. Following the example in that work, if a dataset contains "she picks the wug up in Fresno" and "she puts the wug down in Tempe", then the augmentation method generates samples of puts down substituted into sentences containing picks up. For example the sentence "Pat picks cats up" can be augmented to "Pat puts cats down". GECA relies on being able to identify templates containing discontiguous fragments which contain at least two tokens. In the case of SCAN, GECA might identify the fragment "jump ... JUMP ... JUMP ... JUMP" from the concatenated instruction-action pair "jump thrice JUMP JUMP JUMP" and substitute it into "walk around right thrice WALK RTURN WALK RTURN WALK RTURN" such that it is augmented into "jump around right thrice JUMP RTURN JUMP RTURN JUMP RTURN". As noted by Andreas (2020), the time and space complexity of GECA can be quite large and scales with the number of recognized templates and fragments. In contrast to GECA, meta-seq2seq does not rely on finding templates within the dataset to generate augmentations. Instead, the input and target word-symbol mappings are permuted on every episode, which generates a much wider diversity of data.

The meta-seq2seq architecture is attractive, first because of its strong performance on the sequence generation tasks requiring compositional generalization in SCAN, but more importantly because of its non-domain-specific architecture. It only requires permutation of token-symbol mappings and does not require any domain specific information for generating new sentences. It was shown in Lake (2019) that the compositional generalization can be achieved if the context tasks resemble the query task.

However, it has until now remained an open question how such an approach could be applied to state-conditional problems (Ruis et al., 2020). The main challenge is that both meta-seq2seq rely on *retrieval* from the training data for either their supports or fragments used to generate additional training data. As observed by Mitchell et al. (2021), this is problematic because the supports need to be relevant to the task at hand and they will not be relevant at the point where there is a mismatch between the support state and the query state. We hypothesize that the supports have to be from the same state, which means that they should be generated as such.

The architecture we use in this work extends on Lake (2019) in two ways. The first is a general modernization of the architecture, replacing RNN encoders and decoders with Transformer counterparts, which we do to make our work more comparable to recent works on gSCAN (Qiu et al., 2021; Sikarwar et al., 2022). The second, and more critical extension, is to *generate* supports in the *same state* for each meta-training episode. In this work, we do that through the use of an *oracle function* (example shown in Figure 2), which generates support token instructions and actions by replacing verbs and adverbs with other valid combinations seen in the training data, then generating the corresponding support targets for the new instruction in that environment using the generation procedure provided by Ruis et al. (2020). Support instructions which are in Split H are never generated by the oracle. It is possible that a *query* with the same symbols for `pull ... while spinning` is generated after permutation during training, however the probability of this happening is low. We measured that for a single pass through the training data, approximately 3% of permuted query instructions matched `pull ... while spinning`, 0.3% of the permuted query targets matched `PULL` actions followed by four `LTURN` instructions, and their intersection was 0.001% of the data.

We believe that this architecture is also simple and intuitive to understand through the lens of how humans might think about compositional problems. When faced with an unfamiliar instruction ($I^Q$), the agent thinks of similar instructions ($I_1, ..., I_n$) and their solutions ($A_1, ..., A_n$) in the same environment state. The agent then thinks about how those solutions can be composed in light of the current instruction.

## 4 EXPERIMENTS

We ran experiments to determine the performance of our approach. The Transformer blocks use an embedding size ($d_{\text{model}}$) of 128 units and fully-connected layer size ($d_{\text{FF}}$) of 512 units is used. We use 16 layers for each of the state/instruction Transformer (T), 8 layers for the action supports Transformer Encoder (TE) and and 8 layers for the Transformer Decoder (TD), about 13.2M parameters. The learning rate is $10^{-5}$, we have an effective batch size of 4096, and training iteration count of 30,000. During training, dropout is not used and weight decay is set to $10^{-3}$ with the AdamW optimizer. Beta values are left at their defaults, $\beta_1 = 0.9$ and $\beta_2 = 0.999$. Learning rate warmup is used up to step 5000 to a peak learning rate of $10^{-5}$, then decayed on a log-linear schedule from steps 5000 to 50000 to $10^{-6}$. Gradient norms are clipped at 0.2 to improve training stability. We use 16-bit precision during training and make use of gradient accumulation in order to simulate large batch sizes where memory is limited.

Runs are over 10 seeds and the bottom 3 seeds are excluded by ranking maximum performance on the in-distribution Split A. Since the other splits are in-principle unseen, we show performance on splits B-H by taking a checkpoint for each seed at its best Split A performance, then measuring performance on all the other splits at that checkpoint, shown in Ours(o, A). For completeness we also show the performance on all splits on a checkpoint at the best Split H performance at Ours(o).

Table 2: Our approach compared to other recent works on the gSCAN dataset. Numbers are mean success rate over 10 seeds ± standard deviation. Additional comparisons, including with the original baseline (Ruis et al., 2020), GECA (Andreas, 2020), LCGN (Gao et al., 2020) and Recursive Decoding (Setzler et al., 2022) can be found Appendix B. Not that splits B-F are not directly comparable due to the way that we generate supports, see the later analysis in this work.

| | ViLBERT (Qiu et al., 2021) | Modular (Ruis & Lake, 2022) | Role-guided (Kuo et al., 2021) | Transformer Ours | Ours(o, A) Ours | Ours(o) Ours |
|---|---|---|---|---|---|---|
| #params | 3M | | | 13.2M | 13.2M | 13.2M |
| A | 99.95 ± 0.02 | 96.34 ± 0.28 | 96.73 ± 0.58 | **1.0 ± 0.0** | 0.96 ± 0.0 | 0.95 ± 0.01 |
| B | **99.90 ± 0.06** | 59.66 ± 23.76 | 94.91 ± 1.30 | 0.91 ± 0.15 | 0.96 ± 0.01 | 0.96 ± 0.01 |
| C | **99.25 ± 0.91** | 32.09 ± 9.79 | 67.72 ± 10.83 | 0.86 ± 0.19 | 0.97 ± 0.01 | 0.97 ± 0.01 |
| D | 0.0 ± 0.0 | 0.0 ± 0.0 | 11.52 ± 8.18 | 0.0 ± 0.0 | **0.35 ± 0.06** | **0.35 ± 0.05** |
| E | **99.02 ± 1.16** | 49.34 ± 11.60 | 76.83 ± 2.32 | 0.79 ± 0.26 | 0.98 ± 0.01 | 0.97 ± 0.01 |
| F | 99.98 ± 0.01 | 94.16 ± 1.25 | 98.67 ± 0.05 | **1.0 ± 0.0** | 0.97 ± 0.01 | 0.97 ± 0.01 |
| H | 22.16 ± 0.01 | 76.84 ± 26.94 | 20.98 ± 1.38 | 0.08 ± 0.09 | 0.81 ± 0.03 | **0.86 ± 0.02** |

We include three recent related works for comparison. ViLBERT (Qiu et al., 2021) is general architecture and currently the state-of-the-art on many of the gSCAN splits. It uses a convolutional encoder for the states, cross-attention with a Transformer-like model between instructions and states, then uses this as context for a Transformer decoder to generate actions autoregressively.

"Modular" is a recent work by Ruis & Lake (2022). It uses a specialized decomposition into Perception, Interaction, Navigation and Transformation Modules, each of which are trained independently with their own training targets, then connected together at test time. The modular decomposition gives a prior on how the problem should be solved (for example by decomposition into egocentric and allocentric plans). The work also describes how data augmentation can be used to improve the model, but we show the results coming from use of the modular architecture alone. This approach can get good performance on Splits G and H. Performance on other splits is either slightly improved or comparable to the baseline in Ruis et al. (2020), which is likely due to the use of a similar underlying architecture of RNNs and CNNs as feature encoders.

"Role-guided" (Kuo et al., 2021) uses linguistic priors to decompose the parsing problem and specify how sub-parsers are connected. It can achieve some level of performance on Split D and comparable performance on Split H to the Transformer.

"Transformer" is a baseline Transformer model, different and more vanilla than the ViLBERT in Qiu et al. (2021). The Transformer follows an encoder-decoder structure, where the state and instruction are jointly encoded and the actions are decoded with causal masking. The Transformer uses 28 layers on both the encoder and the decoder, in order to match the parameter count of the meta-seq2seq model. At the same parameter budget and with similar training hyperparameters as meta-seq2seq, it is still not able to solve Split H.

The results are shown in Table 2. We achieve very strong performance on Split H, comparable to the recent work of Ruis & Lake (2022). We provide an analysis of the remaining failure cases in Section 4.2.

There are also a few observations to be made about the results that we report. The first is that the results on Splits B to F, the results are not exactly comparable, because a path to the goal object is indirectly shown through the supports. For example in split C, the challenge might be to "walk" to the unseen "red square" and the supports show a series of actions which satisfy "push" the "red square" or "walk" to the "red square while spinning". Its important to note that the indirect leak does not reveal the true solution, only components which could be composed together into the real solution. The leak also does not happen during training; only during inference on the test set. In practice, we show in Section 3 that the indirect leak does not mean much for practical applicability since the support actions can be generated by a Transformer trained only on the training set. In that light, retained high performance on these splits is impressive, since the model can perform a form of few-shot compositional meta-learning based on generated support targets. The second observation is that we do not consider Split G (the "few-shot-learning" split) our work. Initial experiments showed that our approach does not solve this task when the number of related examples is small. This might seem surprising for a meta-learning architecture, but we believe the reason for this is that Split G requires RTURN, LTURN(3) RTURN after each step, and for any two target actions indices $a^{(1)}$ and $a^{(2)}$, it is not likely that you will see the sequence $a^{(1)}a^{(2)}(3)a^{(1)}$ in the training data. There is a good discussion by Ruis & Lake (2022) about a data-augmentation method which can help to solve this split. The final observation is that while the results reported by Ruis et al. (2020) when using GECA are reproduced in Appendix B, the comparison may not be a fair one, since the underlying baseline architecture is quite different (RNNs as opposed to Transformers) and GECA was only applied to the instructions and state and not to the target commands like we apply permutations for meta-seq2seq.

## 4.1 RELAXING ASSUMPTIONS ON THE ORACLE

The assumption that an oracle is able to generate *perfect* examples of *related* instructions and actions in the same environment state is a strong one, so we also provide some analysis of how extending meta-seq2seq in this way performs when these assumptions are relaxed. We also want to check if the oracle behaviour that we rely on is important for the agent's ability to perform well on Split H. Success rates for a similar experimental setup are shown in Table 3.

**No Permutations**   When we remove the Permuter blocks in our architecture, performance on Splits D and H drop to become are comparable to a Transformer. This confirms our hypothesis about overfitting to particular sequences of symbols and also a similar result in (Lake, 2019).

**Distractor supports**   Ideally we also want to be robust to irrelevant distractor supports if we are to replace the oracle in the future with a generative model that generates a superset of the relevant instructions. Irrelevant distractor supports are those which refer to a non-target object, or to an object which does not exist. We perform the same meta-learning experiment, but this time the oracle generates three additional distractor instructions. In this case, performance is slightly worse, but this is mainly due to higher variance in convergence rate between seeds.

**Irrelevant Instructions**   When adjusting the oracle to only generate irrelevant supports, performance on all splits drops significantly. This is consistent with the analysis of Mitchell et al. (2021).

**Retrieval**   We also tested generating relevant instructions and making demonstrations on different state to the query state by retrieving a state for the query instruction from the training data. The states are included as part of the support inputs. Performance in this case also drops.

**Support Targets generated by Transformer (Transformer Actions)**   Since our ultimate aim is to replace the oracle with a generative model in future work, we also tested a scenario the action supports come from a baseline Transformer which receives the current state and the support instructions generated by the oracle function. In this case the Transformer was trained on the gSCAN training set and had a 95% autoregressive generation success rate on Split A. In this scenario we are no longer doing "few show compositional meta-learning" for Splits A-F, but instead *zero-shot*

Table 3: Different types of oracle behaviour. Numbers are success rates ± standard deviation over top 3 of 10 seeds by Split A performance, Ablations measured at 30,000 iterations at the best validation checkpoint on Split A.

|   | Ours(o, A) | No permutations | Transformer Actions | Distractors | Irrelevant Instructions | Retrieval |
|---|---|---|---|---|---|---|
| A | 0.96 ± 0.0 | 0.97 ± 0.0 | 0.96 ± 0.0 | 0.92 ± 0.05 | 0.27 ± 0.01 | 0.28 ± 0.02 |
| B | 0.96 ± 0.01 | 0.98 ± 0.0 | 0.96 ± 0.0 | 0.94 ± 0.04 | 0.27 ± 0.02 | 0.2 ± 0.03 |
| C | 0.97 ± 0.01 | 0.98 ± 0.0 | 0.97 ± 0.01 | 0.93 ± 0.04 | 0.27 ± 0.0 | 0.01 ± 0.0 |
| D | 0.35 ± 0.06 | 0.03 ± 0.03 | 0.0 ± 0.0 | 0.25 ± 0.1 | 0.02 ± 0.02 | 0.0 ± 0.0 |
| E | 0.98 ± 0.01 | 0.98 ± 0.0 | 0.97 ± 0.01 | 0.94 ± 0.06 | 0.28 ± 0.02 | 0.04 ± 0.02 |
| F | 0.97 ± 0.01 | 0.99 ± 0.0 | 0.97 ± 0.0 | 0.92 ± 0.06 | 0.23 ± 0.02 | 0.37 ± 0.02 |
| H | 0.81 ± 0.03 | 0.16 ± 0.07 | 0.82 ± 0.03 | 0.74 ± 0.09 | 0.0 ± 0.0 | 0.08 ± 0.02 |

generalization. We can still get decent performance on all splits except Split D. This is because the Transformer cannot predict any support sequence to reach an object south-west of the agent, whereas in for other splits, the Transformer can predict a path to the previously unseen object in the context of the instruction. Notably, we match the performance of the oracle-generated actions on Split H.

## 4.2 FAILURE CASE ANALYSIS

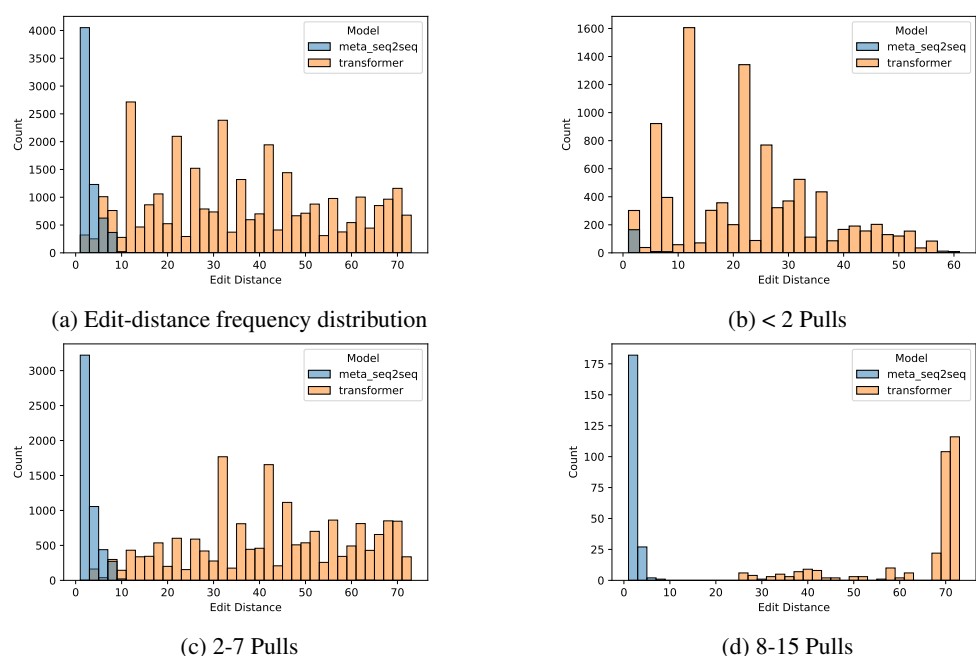

(a) Edit-distance frequency distribution

(b) < 2 Pulls

(c) 2-7 Pulls

(d) 8-15 Pulls

Figure 3: Failure case analysis. In (a), we show the edit distance frequency distribution over all failures. In (b), (c) and (d), we show the edit distance distribution as well as edit distance as a function of the number of PULL instructions in the target sequence. Models that generalize poorly will have a larger edit distance for more complex target instructions

We investigated whether the remaining failures on Split H were due our approach being unable to solve the fundamental generalization problem in Split H or due to some other issue. To diagnose this, we examined edit distances between ground truth sequences in Split H and sequences generated by Ours(o) and the Transformer when run without teacher forcing on the failed examples. If the edit distance length scales with the number of PULL and LTURN(4) combinations required, then it means that the model is likely either generating one of the two instructions, but not both in an interleaved sequence. For example, it might generate actions corresponding to the neighbours 'push while spinning' or 'pull hesitantly'. In practice we found that the edit distances for Ours(o) follow a power law, whereas for a Transformer they scale with the instruction complexity, shown in Figure 3. In the majority of cases, only a small number (1 to 3) errors are made throughout the whole sequence,

and only in a very small number of cases do we make a large number of errors. The edit distance does not scale with the complexity of the target instruction. This indicates that Ours(o) correctly generalizes to the expected behaviour, but makes some other error, different from the Transformer which does not generalize to the expected behaviour and has a edit distance distribution that scales with the target complexity.

Over the same remaining failure cases, we observed four types of failure:

- **Did not turn (78.82%)** The agent "missed" a turn instruction when generating an instruction path that requires one, for example, because the target object is not in the same row as the agent. In this case, `WALK WALK` is generated as opposed to `LTURN WALK` or `RTURN WALK`.

- **Spurious Pull (33.1%)** The agent generates a `PULL` instruction where it should not generate one.

- **Missed Pull (8.09%)** The agent does not generate a `PULL` instruction where it should generate one.

- **Other reason (0.058%)** The failure is more complex or multi-faceted than can be attributed to the above reasons.

Note that multiple failures can happen in a single example, so percentages do not add to 100.

The majority of failures can be attributed to "**Did not turn**". To compute whether the agent is erroneously picking `WALK` but is uncertain, we compute the mean entropy of the prediction logits in all cases where there is a failure to turn. A high amount of uncertainty should correspond to a value of 0.5. The mean value of the entropy plus-minus standard deviation in this case is $0.22 \pm 0.24$. This indicates that the agent is somewhat certain about its decision, but there are cases where it is completely certain (in error) or quite uncertain, where further training may improve the situation.

There are also a few cases where the agent generates a `PULL` instruction or does not generate one when it is expected to ("**Spurious Pull**" and "**Missed Pull**"). We hypothesized that this may be because of an asymmetry between the actions seen for "push while spinning" in the context and "pull while spinning" in the target (the number of `PUSH` or `PULL` actions can differ depending the location of the target object and its surroundings), but we did not see any concrete relationship here.

## 5    DISCUSSION AND CONCLUSION

The extension of meta-seq2seq we present in this paper has promising results in the challenging Split H of the gSCAN benchmark. Split D and G remain challenging to solve with a general approach that does not rely on data augmentation or a problem-specific architecture. There is also the splits proposed in (Qiu et al., 2021) which require *relative* reasoning about objects, where a Transformer can only solve half of the test examples. Finally there are also the test splits in ReaSCAN (Wu et al., 2021) which are comprised of very complex instructions requiring multiple steps of reasoning to find the target object. See Sikarwar et al. (2022) for recent progress on these splits.

At present, an important limitation of this work is that an oracle function generates $I_1, ..., I_n$ and $A_1, ..., A_n$ for a given $I^Q$. We hope to extend this work by replacing the oracle with a generative model, which generates in-distribution query instructions and their corresponding actions with reference to the generative distribution of the training data. There is also a slight degradation of performance on the other splits, including the in-distribution Split A, compared to a Transformer. In light of the limitations, we suggest that applications use this work in conjunction with a baseline, for example by distillation (Hinton et al., 2015) or by using this approach as a sort of "system-2" fallback for when a "system-1" model is uncertain about its inputs (Goyal & Bengio, 2020).

Generalization to unseen instruction compositions remains a challenging problem. Our hypothesis was that meta-seq2seq is a promising general approach and could be extended to grounded language scenarios by an agent which generates a relevant meta-learning context. Our preliminary results show that such an extension has promise and is an area for future work.

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

## A  COMPUTATIONAL RESOURCE USAGE AND REPRODUCIBILITY REQUIREMENTS

Experiments were run on our internal GPU cluster. Running a meta-learning experiment to 30,000 iterations takes about 3 days on a NVIDIA Tesla V100 GPU. For 7 different experiment runs with 10 seeds each, the total compute time is about 210 GPU-days, though in practice the experiments can be run in parallel.

The batch size (4096) we use is quite large and does not fit in GPU memory for a consumer grade GPU. In order to achieve these batch sizes, we use gradient accumulation, so the batch size for each backward step might be 256, but then gradients are averaged over 16 steps to make an effective batch size of 4096. This trades training time for optimization stability.

## B  ADDITIONAL COMPARISONS

We show additional related work comparisons and hyperparameters in Tables 4 and 5.

|   | seq2seq | GECA | FiLM | RelNet | LCGN | Planning | RD Random/RL | Ours(o) |
|---|---------|------|------|--------|------|----------|--------------|---------|
|   | (Ruis et al., 2020) | (Ruis et al., 2020) | (Qiu et al., 2021) | 2021 | (Gao et al., 2020) | 2020 | (Setzler et al., 2022) | Ours |
| A | 97.15 ± 0.46 | 87.6 ± 1.19 | 98.83 ± 0.32 | 97.38 ± 0.33 | 98.6 ± 0.9 | 94.19 ± 0.71 | 98.39 ± 0.17 | 0.95 ± 0.01 |
| B | 30.05 ± 26.76 | 34.92 ± 39.30 | 94.04 ± 7.41 | 49.44 ± 8.19 | 99.08 ± 0.69 | 87.31 ± 4.38 | 62.19 ± 24.08 | 0.96 ± 0.01 |
| C | 29.79 ± 17.70 | 78.77 ± 6.63 | 60.12 ± 8.81 | 19.92 ± 9.84 | 80.31 ± 24.51 | 81.07 ± 10.12 | 56.52 ± 29.70 | 0.97 ± 0.01 |
| D | 0.00 ± 0.00 | 0.00 ± 0.00 | 0.00 ± 0.00 | 0.00 ± 0.00 | 0.16 ± 0.12 |  | 43.60 ± 6.05 | 0.35 ± 0.05 |
| E | 37.25 ± 2.85 | 33.19 ± 3.69 | 31.64 ± 1.04 | 42.17 ± 6.22 | 87.32 ± 27.38 | 52.8 ± 9.96 | 53.89 ± 5.39 | 0.85 ± 0.20 |
| F | 94.16 ± 1.25 | 85.99 ± 0.85 | 86.45 ± 6.67 | 96.59 ± 0.94 | 99.33 ± 0.46 |  | 95.74 ± 0.75 | 0.97 ± 0.01 |
| H | 19.04 ± 4.08 | 11.83 ± 0.31 | 11.71 ± 2.34 | 18.26 ± 1.24 | 33.6 ± 20.81 |  | 21.95 ± 0.03 | 0.86 ± 0.02 |

Table 4: Additional related work comparisons.

|  | ViLBERT | Modular | Role-guided | Transformer | Ours(o, A) | Ours(o) |
|--|---------|---------|-------------|-------------|------------|---------|
|  | (Qiu et al., 2021) | (Ruis & Lake, 2022) | (Kuo et al., 2021) | Ours | Ours | Ours |
| Learning Rate | 0.0015 | 0.001 | 0.001 | 0.0001 | 0.0001 | 0.0001 |
| Batch Size | 128 | 200 | 200 | 4096 | 4096 | 4096 |
| Steps | 114.96K | 73K | 150K | 35K | 35K | 35K |
| #params | 3M |  |  | 13.2M | 13.2M | 13.2M |

Table 5: Hyperparameters used in the related work comparisons of Table 2

## C  ORACLE FUNCTION

| Environment | Query Instructions | Target Actions | Supports | |
|---|---|---|---|---|
|  | walk to a small circle hesitantly | LTURN(2) WALK STAY WALK STAY WALK STAY WALK STAY RTURN WALK STAY | push a small circle hesitantly | LTURN LTURN WALK STAY WALK STAY WALK STAY WALK STAY RTURN WALK STAY PUSH STAY PUSH STAY PUSH STAY PUSH STAY |
| | | | pull a small circle hesitantly | LTURN LTURN WALK STAY WALK STAY WALK STAY WALK STAY RTURN WALK STAY PULL STAY PULL STAY PULL STAY PULL STAY |
| | | | walk to a small circle while spinning | LTURN(4) LTURN LTURN WALK LTURN(4) WALK LTURN(4) WALK LTURN(4) WALK LTURN(4) RTURN WALK |
| | | | walk to a small circle while zigzagging | LTURN LTURN WALK RTURN WALK LTURN WALK WALK WALK |
| | | | walk to a small circle | LTURN LTURN WALK WALK WALK WALK RTURN WALK |
|  | push a red small cylinder while zigzagging | WALK RTURN WALK LTURN WALK RTURN WALK LTURN WALK PUSH | walk to a red small cylinder while zigzagging | WALK RTURN WALK LTURN WALK RTURN WALK LTURN WALK |
| | | | pull a red small cylinder while zigzagging | WALK RTURN WALK LTURN WALK RTURN WALK LTURN WALK |
| | | | push a red small cylinder while spinning | LTURN(4) WALK LTURN(4) WALK LTURN(4) WALK LTURN(4) RTURN WALK LTURN(4) WALK |
| | | | push a red small cylinder hesitantly | WALK STAY WALK STAY WALK STAY RTURN WALK STAY WALK STAY |
| | | | push a red small cylinder | WALK WALK WALK RTURN WALK WALK |

Table 6: Examples of what supports the oracle function generates for a given query instruction and environment state. These two examples are from the training data. Note that we never generate the same instruction as the query instruction in the supports, and we also never generate any Split H instruction in the supports. Also note that in some cases, the environment makes pushing or pulling an object impossible, even though it is in the instruction, see the second row for an example of this.

The oracle function generates relevant instructions by the use of a templating mechanism, which replaces verbs and adverbs in the sentence with other verbs and adverbs, such that the whole combination is still in distribution, but not the same as the query instruction. The rules of the system are:

- Replace "pull" with "push" and "walk to"
- Replace "walk to" with "push" and "pull" (but not if "while spinning" is the adverb)
- Replace "push" with "walk to" and "pull" (but not if "while spinning" is the adverb)
- Replace "while zigzagging" with "hesitantly", nothing and "while spinning" (but not if "push" is the verb)

- Replace "hesitantly" with "while zigzagging", nothing and "while spinning" (but not if "push" is the verb)
- Replace "while spinning" with "hesitantly", "while zigzagging" and nothing

Examples of what the oracle function generates for a given query instruction and environment can be found in Table 6.

Actions are generated by using the same procedure provided in Ruis et al. (2020). The instruction generated by the oracle is given to the demonstration generation procedure and a demonstration is generated by that. A demonstration can also be generated by providing the oracle-generated instruction and current state representation as input to a Transformer model trained on the provided training set.

## D    DICTIONARIES

| Word | Symbol | Action | Symbol |
|---|---|---|---|
| 'a' | 0 | PULL | 0 |
| 'big' | 1 | PUSH | 1 |
| 'blue' | 2 | STAY | 2 |
| 'cautiously' | 3 | LTURN | 3 |
| 'circle' | 4 | RTURN | 4 |
| 'cylinder' | 5 | WALK | 5 |
| 'green' | 6 | | |
| 'hesitantly' | 7 | | |
| 'pull' | 8 | | |
| 'push | 9 | | |
| 'red' | 10 | | |
| 'small' | 11 | | |
| 'square' | 12 | | |
| 'to' | 13 | | |
| 'walk' | 14 | | |
| 'while spinning' | 15 | | |
| 'while zigzagging' | 16 | | |

Table 7: Default mapping of words and actions to symbols

# E    PERMUTER BLOCKS

| Original words | Permutation | Encoded words | Permuted encoding |
|---|---|---|---|
| walk to a blue small cylinder | a(0) → 3, big(1) → 17, blue(2) → 0, cautiously(3) → 16, circle(4) → 8, cylinder(5) → 11, green(6) → 10, hesitantly(7) → 15, pull(8) → 13, push(9) → 4, red(10) → 6, small(11) → 5, square(12) → 12, to(13) → 14, walk(14) → 2, while spinning(15) → 9, while zigzagging(16) → 1, yellow(17) → 7, | 14 13 0 2 11 5 | 2 14 3 0 5 11 |
| pull a green cylinder | a(0) → 7, big(1) → 13, blue(2) → 4, cautiously(3) → 15, circle(4) → 6, cylinder(5) → 5, green(6) → 11, hesitantly(7) → 1, pull(8) → 17, push(9) → 14, red(10) → 12, small(11) → 0, square(12) → 10, to(13) → 3, walk(14) → 8, while spinning(15) → 2, while zigzagging(16) → 16, yellow(17) → 9, | 8 0 6 5 | 17 7 11 5 |
| push a green small square while spinning | a(0) → 0, big(1) → 14, blue(2) → 9, cautiously(3) → 1, circle(4) → 10, cylinder(5) → 17, green(6) → 7, hesitantly(7) → 12, pull(8) → 2, push(9) → 16, red(10) → 15, small(11) → 13, square(12) → 4, to(13) → 6, walk(14) → 8, while spinning(15) → 3, while zigzagging(16) → 11, yellow(17) → 5, | 9 0 6 11 12 15 | 16 0 7 13 4 3 |
| pull a yellow small cylinder hesitantly | a(0) → 6, big(1) → 14, blue(2) → 17, cautiously(3) → 2, circle(4) → 16, cylinder(5) → 0, green(6) → 7, hesitantly(7) → 9, pull(8) → 15, push(9) → 8, red(10) → 10, small(11) → 12, square(12) → 3, to(13) → 1, walk(14) → 11, while spinning(15) → 5, while zigzagging(16) → 13, yellow(17) → 4, | 8 0 17 11 5 7 | 15 6 4 12 0 9 |
| push a big cylinder while spinning | a(0) → 17, big(1) → 0, blue(2) → 4, cautiously(3) → 14, circle(4) → 2, cylinder(5) → 3, green(6) → 9, hesitantly(7) → 16, pull(8) → 7, push(9) → 8, red(10) → 15, small(11) → 11, square(12) → 13, to(13) → 10, walk(14) → 12, while spinning(15) → 6, while zigzagging(16) → 5, yellow(17) → 1, | 9 0 1 5 15 | 8 17 0 3 6 |

Table 8: Instructions and possible mapping permutations generated by the permuter block.

| Original actions | Permutation | Encoded actions | Permuted encoding |
|---|---|---|---|
| WALK(5) RTURN WALK(5) | PULL(0) → 0, PUSH(1) → 5, STAY(2) → 2, LTURN(3) → 1, RTURN(4) → 3, WALK(5) → 4, | 5(5) 4 5(5) | 4(5) 3 4(5) |
| RTURN WALK(3) | PULL(0) → 0, PUSH(1) → 2, STAY(2) → 3, LTURN(3) → 5, RTURN(4) → 4, WALK(5) → 1, | 4 5(3) | 4 1(3) |
| LTURN(4) WALK LTURN(4) WALK LTURN(5) WALK LTURN(4) WALK LTURN(4) WALK LTURN(4) WALK | PULL(0) → 4, PUSH(1) → 5, STAY(2) → 0, LTURN(3) → 2, RTURN(4) → 3, WALK(5) → 1, | 3(4) 5 3(4) 5 3(5) 5 3(4) 5 3(4) 5 3(4) 5 3(4) 5 | 2(4) 1 2(4) 1 2(5) 1 2(4) 1 2(4) 1 2(4) 1 2(4) 1 |
| LTURN WALK STAY WALK STAY WALK STAY WALK STAY | PULL(0) → 3, PUSH(1) → 0, STAY(2) → 2, LTURN(3) → 5, RTURN(4) → 4, WALK(5) → 1, | 3 5 2 5 2 5 2 5 2 | 5 1 2 1 2 1 2 1 2 |
| LTURN WALK STAY WALK STAY | PULL(0) → 0, PUSH(1) → 3, STAY(2) → 4, LTURN(3) → 5, RTURN(4) → 2, WALK(5) → 1, | 3 5 2 5 2 | 5 1 4 1 4 |
| LTURN(4) WALK LTURN(4) WALK LTURN(4) WALK LTURN(4) RTURN WALK LTURN(4) WALK LTURN(4) WALK LTURN(4) WALK LTURN(4) WALK | PULL(0) → 0, PUSH(1) → 4, STAY(2) → 5, LTURN(3) → 1, RTURN(4) → 3, WALK(5) → 2, | 3(4) 5 3(4) 5 3(4) 5 3(4) 4 5 3(4) 5 3(4) 5 3(4) 5 3(4) 5 | 1(4) 2 1(4) 2 1(4) 2 1(4) 3 2 1(4) 2 1(4) 2 1(4) 2 1(4) 2 |
| LTURN WALK(2) PUSH | PULL(0) → 1, PUSH(1) → 0, STAY(2) → 5, LTURN(3) → 3, RTURN(4) → 4, WALK(5) → 2, | 3 5(2) 1 | 3 2(2) 0 |

Table 9: Actions and possible mapping permutations generated by the permuter block.

The permuter block shuffles the indices mapping words to symbols in the dictionary given in Table 7. Tables 8 and 9 give an example of how the permuted sequences might look to the encoders. Essentially the individual symbols no longer hold any special meaning without reference to the demonstrations, only conditional autoregressive probabilities up to a permutation hold meaning.

