# OpenReview forum: "Meta-learning from demonstrations improves compositional generalization"
_ICLR.cc/2023/Conference — Submitted to ICLR 2023_

### Official Review · Reviewer_TWtT · 2022-10-22

**Confidence:** 4
**Correctness:** 3
**Technical Novelty And Significance:** 2
**Empirical Novelty And Significance:** 3
**Recommendation:** 3

**Clarity, Quality, Novelty And Reproducibility:**

## Questions and Concerns
- Did you try a retrieval based approach to mine support examples instead of using the oracle function?
- Section 4.2: I am a bit unsatisfied by this section. I find the statement that "the issue is not due to a lack of generalization" a bit general as any performance reduction on the test set can be attributed to a lack of generalization. What do the authors mean? What is the problem if it is not "generalization"? What would you conclude from this analysis? Are we happy with a model that solves the H split at 86%? Has the model really learnt to compose the two concepts? I think maybe answering these questions could give a bit more depth to this section.

## Typos and Clarity:
- It would be great if you could first introduce GScan and Meta-Seq2Seq, then go on into describing how you modify the existing approach, then present the results.
- Can you provide examples of the permutations applied to I and A? It is hard to understand what happens from the exposition, the description of the "Permuter block" has not been introduced.
- The support set examples in Figure 2 seem to be all the same, is this an error?
- Section 4: "Each cell in the state S is encoded as a bag-of-words..." I don't understand this sentence, the notion of "state" has not been introduced before, GScan has not been introduced.
- In paragraph "Transformers Actions": "The transformer cannot leak the target location for the south-west object" , no such object was introduced before
- ALRED should be ALFRED in the related works?
- "target target" in the Section 4.2

**Strength And Weaknesses:**

Strengths:
- *Strong performance on split H*: The paper presents a model that show promise in improving performance on split H for GScan. Improvements over recent baselines are promising.

- *Nice ablation study*: The ablation study and the analysis are interesting to understand the failures of the method. I particularly appreciate when a particular problem is studied in detail and ablations are provided.

Weaknesses:
- *Limited clarity*: The paper could be more self-contained: I found it hard to figure out what is going on without re-reading the relevant literature. For example the GScan dataset and the original Meta-Seq2Seq model have not been introduced.

- *Limited scope*: In-depth analysis of a particular generalization failure in only one setting, GScan. A more impactful paper would offer perspective on how the method can be fruitful to solve also other compositional generalization datasets (CFQ? Natural Questions?)

- *Limited novelty*: An application of the Meta-Seq2Seq model from Lake et al. 2019 to GScan. I cannot identify whether there exist an intrinsic modelling novelty brought forward by this work, apart from the design of the oracle function that generates support sets. The oracle function resembles the methodology introduced by GECA.

- *Limited applicability*: Ablation offered the key insight that permutations are important to solve split H, but afaicu, permutations require knowing a lot about the structure of the problem (e.g. that walk maps to I_WALK). Is this realistic for other datasets?


**Summary Of The Paper:**

The paper proposes an analysis of a solution for achieving compositional generalization on the grounded SCAN dataset, in particular on its compositional split "H", i.e. the novel adverb-verb combination setting. The model is required to generalize to creating commands for instructions such as `pull X while spinning`, but has never seen `pull` and `while spinning` together during training. The authors approach this problem inheriting the Meta-Seq2Seq approach from Lake et al. (2019) where a Seq2Seq model is trained to use a batch of support instructions and target commands pairs to predict the output command for a given query instruction. They create the support set using an oracle function (which is actually a manual instantiation of GECA (Andreas, 2020)) and show that permuting the mapping from each instruction to the command (not clear how the permutation was done, but I can imagine by reading the Meta-Seq2Seq paper). They show that the resulting model can outperform the baselines on split H (and D) and provide an ablation of the most important components, namely the use of permutations and the oracle function to create support examples.


**Summary Of The Review:**

I do overall appreciate the relevance of the problem the paper is trying to solve and the ablations therein. Lack of novelty and limited scope indicate this paper might be more suited for a workshop. In its present form, the paper lacks clarity, it is not self-contained and thus it might not be ready for publication just yet.

---

> ### Author Response · Authors · 2022-11-18
> **Thank you for your review!**
>
> We thank the reviewer for the time taken to carefully review our work and for their detailed and useful feedback provided. We have revised the paper in line with the feedback provided and would like to point out the following:
>
> - Did we try a retrieval based approach? We ran experiments to test retrieval for relevant instructions and the results are shown in Table 3 under "Different States". For clarity, the section has now been renamed to “Retrieval”. The results were not good. It is not mentioned in the paper, but we also noticed that validation loss was increasing for all splits under this ablation, so we don't believe that the results will improve with further training. See also the analysis in Mitchell et al. 2021 about the importance of informative examples during meta-learning for generalization.
>
> - GECA and Meta-seq2seq: GECA works by looking at the training data to find co-occurring word pairs and possible substitutions, meta-seq2seq works in a fundamentally different way, it works by permuting the entire vocabulary and using meta-learning to resolve the meaning of the symbols at meta-testing time. Both are reasonable approaches for this sort of problem, although GECA had been tried in Ruis et al. 2020 without a great deal of success.
>
> - We have revised Section 3.2 to improve the clarity of the method and have tried to make the paper more self-contained. We have explained the background behind the meta-seq2seq method and explained the key terminology and how the pieces fit together.
>
> - We have included new appendices D and E to specify the token/symbol mapping and also illustrate the operation of the Permuter blocks.
>
> - We have fixed the typo in Figure 2 where the same instruction was repeated many times. Thank you for pointing this out!
>
> - We have re-structured Section 4.2, clearly specifying the research question we sought to answer. We analyze the remaining errors made on Split H by the proposed approach and conclude that the trajectories proposed by our method are very close to the expected trajectories as compared to a Transformer. The errors are in one or two tokens, which suggests that you do not have a fundamental failure of generalization and that if you train the method for longer that the results might improve further.
>
> - On scope and applicability: Please see the meta-comment.

---

### Official Review · Reviewer_wsCX · 2022-10-24

**Confidence:** 3
**Correctness:** 3
**Technical Novelty And Significance:** 2
**Empirical Novelty And Significance:** 2
**Recommendation:** 3

**Clarity, Quality, Novelty And Reproducibility:**

- The details of  “oracle templating function” lacks from at least of the main paper. I'm not confident readers can reproduce their results from these writings.

- As a note, I didn’t recommend that authors present the statistics of test sets on the paper because I allow readers to know (or even leak) the knowledge of the test set. Although in this paper it is somehow inevitable for discussion.

- As a minor comment, the writing is grammatical, but not so well-composed or fluent. It sometimes lacks clear arguments.

- Typo:
ALRED -> ALFRED


**Strength And Weaknesses:**

- The strength of this paper is that they obtained better performance than the previous models with “oracle templating function” in limited test splits, although there are following problems as written in weakness.

- The first problem is that the proposed model uses the periluminal knowledge of some testsets such as the different distributions of verbs and adverbs mentioned in Sec 3.2. It is prohibited in the previous studies compared in Table 2. The details of the contribution “oracle templating function” is written in Appendix, not in the main paper. It still lacks details.

- The second problem is that the contribution / observation is limited. In this paper, authors proved that tuning Meta-Sequence-to-Sequence learning approach on the limited testset of gSCAN is effective. However, this is not so surprising because some testsets of gSCAN are designed for different test situations. Therefore, we can tune our models for some specific test sets if we preliminary know the information of the testset, such as the different distributions of verbs and adverbs. When we turn some models into the specific distribution, we obviously sacrifice the performance in other test sets with different distributions. This is the observation of the main score Table 2 and is not a novel thing in the learning of the out-of-domain adaptation. Hence the overall paper contribution looks not obscure. In short, optimizing models just for a few good test sets is not a good contribution.

- The third problem is indeed the largest problem: the motivation to access the statistics/bias of gSCAN test set and taking advantage of the existing bias in tuning model is not discussed well in this paper. I’d like to know concrete applications or explanations how the proposed models / approaches for the limited OOD test sets are useful in the following studies or real applications. If the motivation in this paper relies on human compositional problem solving as written in the introduction, how the observations in the experiments, especially for the results of “oracle templating function,” have contributed to it?


**Summary Of The Paper:**

This paper addresses the generalization ability of language-instructed agents of gSCAN. Based on the Meta-Sequence-to-Sequence learning approach and meta-seq2seq architecture of Lake 2019, they issue the statistical action-bias of gSCAN split-H and extend the Meta-Sequence-to-Sequence learning approach to the specially case of split-H. They introduce “oracle templating function” tune the model on the different in-distribution set of split-H. They also provide the model improvements, such as replacing RNNs with Transformers. They succeeded to tune their models to some specific test sets of Split-D and Split-H sacrificing performance in other testsets, using the preliminary known information of the testsets.

Entirely authors include both the contribution of the proposed “oracle templating function” in Sec 3.2 and the comparably minor model improvement (e.g., swapping RNNs with Transformer and minor neural network modification / engineering). It is regrettable that authors mix two parallel contributions and hence it becomes less clear what is the major contribution of the paper. Please clarify the differences from Lake 2019 concisely in the manuscript. In the current paper, authors raised many incremental improvements one by one in Sec 3.2 seemingly irreverent to the main contribution of “oracle templating function.” The details and insights of “oracle templating function” are not clear. Overall, the arguments of this paper are not still convincing yet.

**Summary Of The Review:**

Overall, the paper arguments are still not clear nor convincing yet. The motivation doesn't sound. I hope authors have further discussion on the motivation and the next directions of the entire paper possibly in other conferences or workshops.

---

> ### Author Response · Authors · 2022-11-18
> **Thank you for your review!**
>
> We thank the reviewer for the time taken to carefully review our work and for their detailed and useful feedback provided. We have revised the paper in line with the feedback provided and would like to point out the following:
>
> - We have revised Section 3.2 to more clearly delineate the nature of our contributions, with an emphasis on the importance of generating relevant support instructions using the oracle function. We have also provided some further explanation as to the architectural modernization (for example, using Transformer encoders as opposed to RNNs). This was mainly for the purpose of comparability with other recent works. We do not claim that this is a major contribution of our work.
>
> - On the generality of our model: We would like to point out that in our work, we show that the method does not substantially impact performance on other splits. We believe that proposing a method to address unsolved splits on gSCAN is important because of their relevance to the field of grounded language learning more generally. See also our comments on applications in the meta-comment.
>
> - On the use of test set statistics: The test sets are included with the gSCAN dataset and are intended to be public. The nature of these test sets and the challenges posed by them are also discussed in (Ruis et al. 2020 and Qiu et al. 2021).  We believe it is reasonable to discuss their statistics, because they are out-of-distribution tests and discussing their statistics shows the nature of how the model will need to generalize.
>
> - We have revised the paper to more clearly specify the operation of the oracle function and how it fits into our overall architecture.

---

### Official Review · Reviewer_xpEo · 2022-10-25

**Confidence:** 2
**Clarity, Quality, Novelty And Reproducibility:** please see above
**Correctness:** 4
**Technical Novelty And Significance:** 2
**Empirical Novelty And Significance:** 2
**Recommendation:** 3

**Strength And Weaknesses:**

Strength

They study an important problem.

Using meta-learning for compositional generalization is reasonable.

The results on Split H are positive and they also conducted a range of ablation and error analysis.

Weakness

The paper is poorly organized and very hard to follow. This is the key weakness, but this makes me find it really difficult to judge the overall technical quality and significance.

E.g., after reading the related sections a few times, I still do not understand how meta-seq2seq is applied in their setting and how the technical components handle the gSCAN environments. It seems that many important introductions and remarks are missing. So I suggest that the authors give a technical introduction of the framework and more precisely discuss how it can solve the problem of interest, possibly with visual illustrations.

Some important concepts are repeatedly used without a definition. E.g., what is a "support"? The paper has "support set" and "support instructions" at many places but it is unclear to me what it actually means. I also read the original gSCAN paper but they didn't use this term at all. Similarly, what is an "oracle"? What is "in-distribution" instructions and how would an instruction from non-oracle look to an instruction from oracle?



**Summary Of The Paper:**

This paper works on compositional generalization of language-instructed agents.
They apply meta-seq2seq method to this setting.
They conduct experiments on the gSCAN benchmark.

**Summary Of The Review:**

The paper aims to address an interesting problem and their general technical idea (using meta-learning) is reasonable.

But the paper is poorly written and it is hard to give a good judgement on its current version.

---

> ### Author Response · Authors · 2022-11-18
> **Thank you for your review!**
>
> We thank the reviewer for the time taken to carefully review our work and for their detailed and useful feedback provided. We have revised the paper in line with the feedback provided and would like to point out the following:
>
> - We have made significant revisions to the description of the Meta-Sequence-to-Sequence and oracle support generation methods described in Section 3.2, with the aims of making the paper more self-contained, adding definitions of terms used throughout the paper such as “supports” and “oracle”, describing the exact nature of the problem that solve compared to prior work and more clearly delineating our own contributions atop of Lake 2019.
>
> - On the question of novelty of contributions, please see our meta-comment.

---

### Official Review · Reviewer_EJmn · 2022-10-26

**Confidence:** 4
**Correctness:** 3
**Technical Novelty And Significance:** 2
**Empirical Novelty And Significance:** 3
**Recommendation:** 3

**Clarity, Quality, Novelty And Reproducibility:**

** Clarity **

The paper is generally well written and is a pleasure to read. I especially enjoyed the literature review section which is both thorough and concise.

I was a bit confused about Figure 4. I expected the support instructions to be different, not just a copy of the same support instruction. Maybe choosing a different example would be more insightful.

Also, the authors refer to appendices in the paper, but the paper does not have any (they are included in the supplementary materials instead). Generally appendices are usually included in the paper, while supplementary materials can be used to provide code or even more extra information that does not naturally fit into appendices.

Generally, these are minor points that did not affect my evaluation. Overall, the clarity of the paper satisfies ICLR standards.

** Reproducibility **

Clear descriptions, together with the code that authors provide, make the paper satisfy the highest reproducibility standards.

** Novelty **

The novelty and significance of the paper are, unfortunately, very much borderline. While the paper does show improvements on two splits in gSCAN, the value and the overall impact of this specialized architecture may not be high enough.

Additionally, while the paper does outperform another specialized architecture developed for the gSCAN dataset (Improving Systematic Generalization Through Modularity and Augmentation, Ruis, Lake, 2022), I don't believe that this improvement is sufficiently consequential, since that method is an archive pre-print, and, while providing valuable ideas, does not in itself establish a standard basis for comparison.

The method is, without a doubt, elegant, and it may provide a pathway towards modeling abstraction in language models, but its key idea was already known (Lake 2019). Therefore, we must resort to judging the contribution based on how impactful its applications are, and I am not sure if the impact is sufficient to meet the standards of the ICLR conference.

** Quality **

The experiments fit the goal of the paper and are clearly reported. There is, however, one potential concern I have about the experimental design (see "questions") section below.

** Questions **

The authors say "The oracle never generates an example of the query instruction or instructions from test Split H," but it's not entirely clear whether it accounts for random permutations. I.e. the authors say "critically, we make random permutations of word/symbol assignments for both the instruction and the support commands". Is there a chance that an oracle generated something, then it was permuted, and the result actually does happen to be in the test Split H?


**Strength And Weaknesses:**

Strengths:
- The paper addresses a highly relevant problem
- The proposed method is elegant and might allow to model abstraction and generalization
- The proposed method shows promising results

Weaknesses:
- The method is very closely related to the work "Compositional generalization through meta
sequence-to-sequence learning" by Lake (2019), which it builds upon. While adapting it to transformer architecture and a new task is not trivial, it does limit the conceptual novelty of the contribution.
- Empirical results, while promising, are confined to a relatively narrow domain. This is especially unfortunate, since when novelty is relatively limited, the breadth and impact of empirical results becomes paramount.

**Summary Of The Paper:**

The paper expands the Meta-Sequence-to-Sequence learning approach proposed by B. Lake in 2019, adapting it to be used with a modern transformer architecture and applying it to the gSCAN benchmark, utilizing the idea of random symbol/token permutation as a remedy against overfitting to specific sequences in the training set. The authors show that performance on two challenging splits of the dataset is improved, compared to previous works.

**Summary Of The Review:**

I enjoyed reading the paper which addresses a highly important and ambitious problem of compositional generalization. I believe that the proposed method is interesting and has the potential to be generalized to model abstraction and generalization in a wider array of applications. Unfortunately, in its present state, I feel that the combination of relatively limited novelty with the relatively narrow range of applications puts this paper below the threshold of acceptance.

---

> ### Author Response · Authors · 2022-11-18
> **Thank you for your review!**
>
> We thank the reviewer for the time taken to carefully review our work and for their detailed and useful feedback provided. We have revised the paper in line with the feedback provided and would like to point out the following:
>
> - Appendices have now been included in the main paper
> - We have addressed the specific concern of leaking from Split H into the training data using the Permuter. Examples which could be from Split H are generated with very low probability. We have provided an empirical calculation in Section 3.2 to show that this is likely a negligible proportion of the training data.
> - We have more clearly delineated the extent of our contributions in Section 3.2 atop of Lake et al. 2019. Please see our meta-response for more details.
> - We have more clearly specified the comparison with (Ruis & Lake 2022), in particular indicating to what extent our work is comparable, given the differing underlying architectures used in Ruis & Lake and more recent works such as our own and that of Qiu et al. 2021 and Sikawar et al. 2022.
> - On the question of applications of the work, please see the meta-comment.

---

> > ### Comment · Reviewer_EJmn · 2022-11-23
> > **Thank you for your response**
> >
> > Thank you for your thoughtful response. Unfortunately, the issues raised by me and other reviewers run quite deep, and, I believe, are hard to resolve given the short rebuttal timeframe. I do believe that the paper has potential, but it needs to be demonstrated with more experiments to demonstrate the general importance and reliability of the obtained results. At present, unfortunately, I can not increase my score, but I hope to see an improved version of the paper published one day!

---

### Author Response · Authors · 2022-11-18
**Thank you to the reviewers! Meta-comment on the reviews**

We thank the reviewers for the time taken to carefully review our work and for the detailed and useful feedback provided. We have revised and improved the paper in line with the feedback provided. We have replied to each review to address specific review feedback and we wish to point out the following in relation to common feedback we received.

The reviewers have pointed out the following strengths of the paper:

- Strong performance improvement on Split H.

- Meta-learning being an appropriate approach to solve the problem.

- Detailed ablation study showing the importance of relevant demonstrations when using meta-learning to solve compositional problems in the grounded setting.

The reviewers had the following common concerns, for which we would like to point out the following:

- Novelty: Our main contribution is that we show that meta-seq2seq (Lake 2019) can work in grounded language learning problems, which was flagged as an interesting research question in other works (Ruis et al. 2020). We show that Meta-seq2seq can work in grounded language learning problems as long as the agent is given relevant demonstrations. We show that directly applying the algorithm from (Lake 2019) with the same retrieval based approach (where the agent queries demonstrations from the training set) is not sufficient. What is critical is that relevant demonstrations are provided, eg, those which are in the same state and show the types of behaviours that the agent will need to compose. This is consistent with the analysis by Mitchell et al. 2021 in the ungrounded setting on SCAN. In our work we generate demonstrations such that they are relevant. This is what unlocks performance on difficult compositional generalization splits like split H and is the key contribution of our work. We have clarified the contributions in the revised submission.

- Scope and Applicability: We believe that gSCAN demonstrates challenging cases that language-grounded agents will need to handle. These types of problems will arise in many different grounded language learning scenarios. Understanding limitations of standard approaches and methods to handle them are of paramount importance.

- Clarity: We have revised the description of the method in Section 3.2 in order to make the paper more self-contained and clearly delineate our contributions atop of (Lake 2019).

---

### Decision · Program_Chairs · 2023-01-20

**Decision:**

Reject

**Justification For Why Not Higher Score:**

All the reviewers and I agreed unanimously that this was a clear reject (3). The reasons include: lack of clarity, limited impact, limited novelty, limited evaluation on one dataset, limited success on that dataset.

**Justification For Why Not Lower Score:**

N/A

**Metareview: Summary, Strengths And Weaknesses:**

This paper introduces an adaptation of the meta-learning framework, meta-seq-2-seq, by Lake et al., suitable for The method is evaluated on gScan, a dataset designed to test compositional generalization in grounded language setting.

Compositional generalization is clearly a central problem for efficient machine learning and robust generalization and I think any work in this area is welcome.

All the reviewers agreed that the paper demonstrates strong performance on gScan split H and the approach was interesting and had potential, but was also somewhat incremental over the Lake model, and limited in scope. All were also agreed that the writing needs to be clearer, more self-contained, and referenced results and other material better introduced.

**Summary Of Ac-Reviewer Meeting:**

N/A